# Early prediction of in-hospital death of COVID-19 patients: a machine-learning model based on age, blood analyses, and chest x-ray score

Emirena Garrafa[1,2]*[†], Marika Vezzoli[1†], Marco Ravanelli[3,4], Davide Farina[3,4], Andrea Borghesi[3,4], Stefano Calza[1], Roberto Maroldi[3,4]

[1]Department of Molecular and Translational Medicine, University of Brescia, Brescia, Italy; [2]ASST Spedali Civili di Brescia, Department of Laboratory, Brescia, Italy; [3]Department of Medical and Surgical Specialties, Radiological Sciences and Public Health, University of Brescia, Brescia, Italy; [4]ASST Spedali Civili di Brescia, Department of Radiology, Brescia, Italy

**Abstract** An early-warning model to predict in-hospital mortality on admission of COVID-19 patients at an emergency department (ED) was developed and validated using a machine-learning model. In total, 2782 patients were enrolled between March 2020 and December 2020, including 2106 patients (first wave) and 676 patients (second wave) in the COVID-19 outbreak in Italy. The first-wave patients were divided into two groups with 1474 patients used to train the model, and 632 to validate it. The 676 patients in the second wave were used to test the model. Age, 17 blood analytes, and Brescia chest X-ray score were the variables processed using a random forests classification algorithm to build and validate the model. Receiver operating characteristic (ROC) analysis was used to assess the model performances. A web-based death-risk calculator was implemented and integrated within the Laboratory Information System of the hospital. The final score was constructed by age (the most powerful predictor), blood analytes (the strongest predictors were lactate dehydrogenase, D-dimer, neutrophil/lymphocyte ratio, C-reactive protein, lymphocyte %, ferritin std, and monocyte %), and Brescia chest X-ray score (https://bdbiomed.shinyapps.io/covid-19score/). The areas under the ROC curve obtained for the three groups (training, validating, and testing) were 0.98, 0.83, and 0.78, respectively. The model predicts in-hospital mortality on the basis of data that can be obtained in a short time, directly at the ED on admission. It functions as a web-based calculator, providing a risk score which is easy to interpret. It can be used in the triage process to support the decision on patient allocation.

*For correspondence:
emirena.garrafa@unibs.it

[†]These authors contributed equally to this work

**Competing interest:** The authors declare that no competing interests exist.

## Introduction

Starting from late February 2020, the COVID-19 outbreak struck the north of Italy causing more than 30,000 deaths in Lombardy alone, up to the end of March 2021. At the beginning of the outbreak, the Spedali Civili di Brescia (SCBH), the university hospital of one of the hardest hit cities in Europe, was faced with a 'flash flood' of severely ill patients seeking admission to the emergency department (ED). For several weeks, their number exceeded the available resources, obliging a continuous organizational restructuring of the hospital wards (*Garrafa et al., 2020b*).

In those weeks, given the limited evidence of clinically proven predictors (*Marengoni et al., 2021*; *Wynants et al., 2020*; *Sperrin et al., 2020*), prioritizing hospital admission of non-critical patients was an arduous task. Essentially, the criteria were based on the presence of fever, respiratory symptoms,

and the level of blood oxygenation. A significant drawback of this approach was that patients referring to the ED with very similar clinical findings underwent inconsistent assessments. In this scenario, the availability of predictors would have been extremely beneficial, not only to triage patients, but also to monitor hospitalized patients and warn of exacerbation of the outbreaks.

Starting from March 2020, all patients referred to EDs underwent a chest X-ray at admission or within a few hours. With the purpose of grading pulmonary involvement and tracking changes objectively over time, a chest X-ray severity score was developed (Brescia X-ray score) (*Borghesi and Maroldi, 2020*; *Maroldi et al., 2021*; *Borghesi et al., 2020a*; *Borghesi et al., 2020b*). The score was able to predict in-hospital mortality in 302 patients. In addition to the chest X-ray severity score, a dedicated blood sampling profile was included in the COVID-19 ED work-up (*Garrafa et al., 2020a*). Among its 17 blood analytes, the sampling profile encompassed hemachrome, inflammation biomarkers such as C-reactive protein (CRP), lactate dehydrogenase (LDH), and ferritin, and coagulation markers (fibrinogen and D-dimer). Since that time, the medical literature began to encompass an increasing number of studies advocating the prognostic value of single or grouped blood parameters (*Bonetti et al., 2020*; *Borghi et al., 2020*; *Avouac et al., 2021*; *Knight et al., 2020*). All these parameters were present in our COVID-19 sampling profile.

This study aims to develop and validate an early-warning model (BS-EWM), predictive of in-hospital death, based on data that could easily be acquired on admission to the ED: age, simple blood biomarkers, and chest X-ray. The model was constructed based on the analysis of a cohort of 2872 COVID-19 patients treated in a single reference center over a 10 -month period.

This paper adheres to the TRIPOD checklist for predictive model development and validation (*Collins et al., 2015*).

The study was approved by the local ethics committee (COVID-SURG-BS; NP 4059).

## Results

### Description of the sample

The entire sample analyzed in this paper contained 2782 COVID-19 patients (1010 female [36.3%] and 1772 male [63.7%]), admitted to the ED and hospitalized at SCBH from March to December 2020. During these 10 months, the pandemic had two temporal waves: March–April (MA) (2106 patients, 75.70 % of the entire sample) and May–December (MD) (676 patients, 24.30 % of the entire sample) (*Supplementary file 1a*). The model was trained on a subsample extracted from the first wave (70%) and tested (i) on data not used to calibrate the model (remaining 30 % from the first wave) and (ii) on data from the second wave (*Figure 1*).

The first-wave subsample contained 2106 COVID-19 patients hospitalized in March–April 2020 at SCBH: 744 females (35.3%) and 1362 males (64.7%) (*Table 1*). During that period, 423 patients died (20.09 % of the total): 131 females (31%) and 292 males (69%). Their mean age ± SD was 66.89 ± 14.19: 67.93 ± 15.40 for females and 66.32 ± 13.45 for males (p-value = 0.001). The mean age of deceased patients was 76.21 ± 9.12, while for living patients, it was 64.55 ± 14.27 (p-value < 0.001). Mean hospital stay was 13.58 ± 11.58 days (from a minimum of 0 to a maximum of 140 days): 11.33 ± 10.98 days for patients who died, 14.15 ± 11.66 days for surviving patients (p-value < 0.001).

The second-wave subsample contained 676 COVID-19 patients hospitalized in MD 2020 at SCBH: 266 females (39.3%), 410 males (60.7%) (*Table 1*). During the 8 months of the second wave, 82 patients died (12.13%): 26 females (31.7%) and 56 males (68.3%). The mean age of deceased patients was 76.72 ± 10.79 vs. 65.30 ± 15.20 for surviving patients (p-value < 0.001). The mean hospital stay was 15.35 ± 11.58 days (from a minimum of 0 to a maximum 79 days): 17.77 ± 10.75 days for patients who died, 14.95 ± 11.67 days for surviving patients (p-value = 0.008).

The descriptive statistics for all variables in the dataset are presented in *Supplementary file 1b* and were computed and stratified by the two waves (MA vs. MD) and by outcome (alive vs. dead). The two subsets were similar for most variables.

The correlations between the 17 analytes and the Brescia X-ray score were investigated using Spearman correlation coefficients and visualized using a correlation plot (*Figure 2*). The Brescia X-ray score was positively correlated with neutrophil to lymphocyte ratio (NLR), CRP, LDH, standardized ferritin, and D-dimer, and was negatively correlated with lymphocyte %, monocyte %, and basophil %.

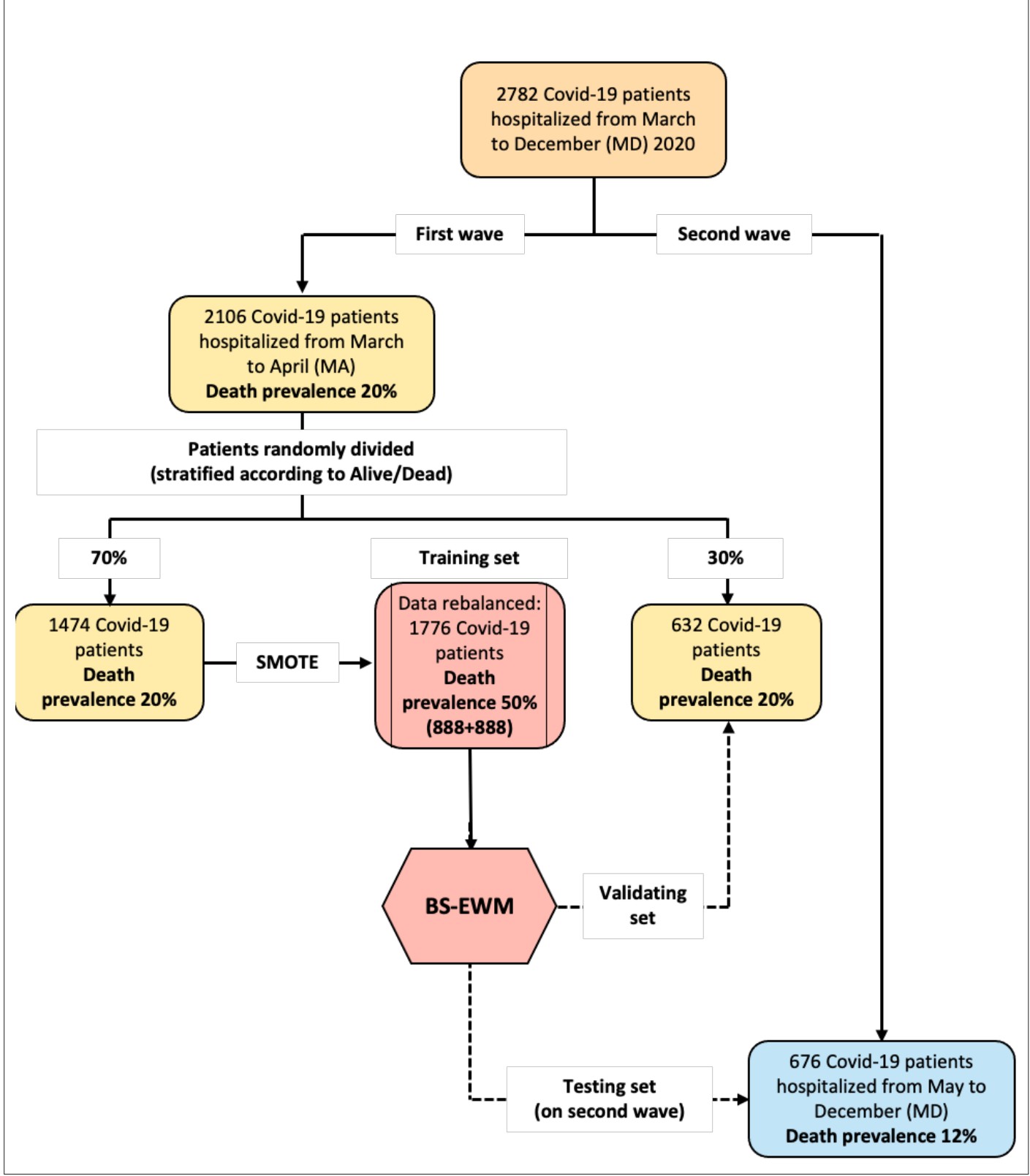

**Figure 1.** Flowchart of the data used in the empirical analyses. The early-warning model (BS-EWM) was trained with a random forest on 70 % of first-wave patients (rebalanced with the synthetic minority oversampling technique [SMOTE] procedure) and (**i**) validated on remaining 30 % of first-wave patients (ii) tested on 676 second-wave patients. In detail, 2106 patients were randomly in training and validating, maintaining the same death prevalence of the first wave.

**Table 1.** Descriptive statistics on all variables in the dataset stratified respect alive–dead. Comparison between first (March–April) and second (May–December) wave.

| Variables | First wave: March–April (MA) 2020 | | | Second wave: May–December (MD) 2020 | | |
| --- | --- | --- | --- | --- | --- | --- |
| | Alive (N = 1683) | Dead (N = 423) | p-Value | Alive (N = 594) | Dead (N = 82) | p-Value |
| **Age** | | | <0.001* | | | <0.001* |
| Mean (SD) | 64.55 (14.27) | 76.21 (9.12) | | 65.30 (15.20) | 76.72 (10.79) | |
| Median (Q1, Q3) | 65.00 (55.00, 75.00) | 77.00 (72.00, 82.00) | | 67.00 (55.00, 77.00) | 80.00 (72.25, 84.75) | |
| Range | 19.00–97.00 | 44.00–98.00 | | 18.00–97.00 | 44.00–98.00 | |
| **Sex** | | | 0.036† | | | 0.131† |
| F | 613 (36.4%) | 131 (31.0%) | | 240 (40.4%) | 26 (31.7%) | |
| M | 1,070 (63.6%) | 292 (69.0%) | | 354 (59.6%) | 56 (68.3%) | |
| **Days in hospital** | | | <0.001* | | | 0.008* |
| N-Miss | 1 | 0 | | 95 | 0 | |
| Mean (SD) | 14.15 (11.66) | 11.33 (10.98) | | 14.95 (11.67) | 17.77 (10.75) | |
| Median (Q1, Q3) | 11.00 (7.00, 18.00) | 8.00 (4.00, 15.00) | | 12.00 (7.00, 20.00) | 17.50 (9.00, 25.00) | |
| Range | 0.00–140.00 | 0.00–88.00 | | 0.00–79.00 | 2.00–46.00 | |
| **Score** | | | <0.001* | | | <0.001* |
| Mean (SD) | 6.92 (4.40) | 8.77 (4.39) | | 5.65 (4.48) | 8.23 (4.63) | |
| Median (Q1, Q3) | 7.00 (3.00, 10.00) | 9.00 (6.00, 12.00) | | 5.00 (2.00, 9.00) | 9.00 (5.25, 11.00) | |
| Range | 0.00–18.00 | 0.00–18.00 | | 0.00–18.00 | 0.00–17.00 | |
| **D-dimer** | | | <0.001* | | | <0.001* |
| N-Miss | 406 | 113 | | 128 | 16 | |

*Table 1 continued on next page*

*Table 1 continued*

| Variables | First wave: March–April (MA) 2020 | | | Second wave: May–December (MD) 2020 | | |
|---|---|---|---|---|---|---|
| | Alive (N = 1683) | Dead (N = 423) | p-Value | Alive (N = 594) | Dead (N = 82) | p-Value |
| Mean (SD) | 1155.03 (2218.51) | 3124.25 (8070.21) | | 1538.17 (3123.38) | 4712.44 (8897.82) | |
| Median (Q1, Q3) | 443.00 (262.00, 985.00) | 944.50 (476.50, 2970.75) | | 739.50 (427.50, 1341.25) | 1112.00 (725.50, 3619.25) | |
| Range | 200.00–47 228.00 | 200.00–60,342.00 | | 190.00–33,501.00 | 190.00–35,000.00 | |
| **Fibrinogen** | | | 0.951* | | | 0.778* |
| N-Miss | 339 | 117 | | 54 | 8 | |
| Mean (SD) | 530.53 (194.13) | 530.55 (213.69) | | 523.94 (169.43) | 519.77 (213.05) | |
| Median (Q1, Q3) | 520.00 (381.00, 650.00) | 515.00 (381.00, 654.00) | | 512.00 (405.00, 612.00) | 510.00 (330.50, 649.00) | |
| Range | 119.00–1339.00 | 68.00–1333.00 | | 147.00–1371.00 | 153.00–1287.00 | |
| **LDH** | | | <0.001* | | | <0.001* |
| N-Miss | 188 | 92 | | 61 | 7 | |
| Mean (SD) | 321.25 (227.50) | 433.71 (205.10) | | 308.30 (196.23) | 443.49 (707.95) | |
| Median (Q1, Q3) | 283.00 (222.00, 373.00) | 406.00 (269.50, 545.50) | | 273.00 (218.00, 354.00) | 332.00 (257.00, 442.50) | |
| Range | 90.00–6689.00 | 123.00–1365.00 | | 108.00–2565.00 | 122.00–6310.00 | |
| **Neutrophils** | | | <0.001* | | | <0.001* |
| N-Miss | 23 | 19 | | 4 | 1 | |
| Mean (SD) | 5.67 (3.61) | 7.17 (4.39) | | 5.80 (3.97) | 7.21 (4.13) | |
| Median (Q1, Q3) | 4.83 (3.29, 7.03) | 6.20 (4.12, 9.02) | | 4.78 (3.42, 7.11) | 6.72 (4.00, 9.77) | |
| Range | 0.00–53.99 | 0.17–30.45 | | 0.10–47.03 | 0.19–23.02 | |
| **Lymphocytes** | | | <0.001* | | | <0.001* |

Table 1 continued

| Variables | First wave: March–April (MA) 2020 | | | Second wave: May–December (MD) 2020 | | |
|---|---|---|---|---|---|---|
| | Alive (N = 1683) | Dead (N = 423) | p-Value | Alive (N = 594) | Dead (N = 82) | p-Value |
| N-Miss | 23 | 19 | | 4 | 1 | |
| Mean (SD) | 1.43 (5.48) | 1.19 (4.29) | | 1.22 (0.81) | 1.38 (4.63) | |
| Median (Q1, Q3) | 1.04 (0.75, 1.42) | 0.81 (0.55, 1.18) | | 1.06 (0.72, 1.52) | 0.74 (0.47, 1.06) | |
| Range | 0.10–177.63 | 0.04–85.51 | | 0.08–10.28 | 0.08–42.20 | |
| **Neutrophils on lymphocytes** | | | <0.001* | | | <0.001* |
| N-Miss | 23 | 19 | | 4 | 1 | |
| Mean (SD) | 6.18 (5.87) | 10.72 (11.71) | | 7.19 (9.92) | 12.84 (13.09) | |
| Median (Q1, Q3) | 4.52 (2.84, 7.50) | 7.13 (4.47, 13.06) | | 4.32 (2.63, 8.40) | 8.50 (4.05, 15.19) | |
| Range | 0.00–101.90 | 0.01–129.67 | | 0.12–143.25 | 0.11–70.56 | |
| **Neutrophils %** | | | <0.001* | | | <0.001* |
| N-Miss | 22 | 19 | | 4 | 1 | |
| Mean (SD) | 0.73 (0.13) | 0.80 (0.12) | | 0.73 (0.13) | 0.79 (0.16) | |
| Median (Q1, Q3) | 0.74 (0.66, 0.82) | 0.82 (0.75, 0.88) | | 0.73 (0.64, 0.83) | 0.83 (0.69, 0.89) | |
| Range | 0.00–0.97 | 0.01–0.97 | | 0.10–0.99 | 0.10–0.96 | |
| **Lymphocytes %** | | | <0.001* | | | <0.001* |
| N-Miss | 22 | 19 | | 4 | 1 | |
| Mean (SD) | 0.18 (0.11) | 0.13 (0.09) | | 0.18 (0.11) | 0.13 (0.13) | |
| Median (Q1, Q3) | 0.16 (0.11, 0.23) | 0.11 (0.07, 0.17) | | 0.17 (0.10, 0.25) | 0.10 (0.06, 0.18) | |
| Range | 0.01–0.97 | 0.01–0.99 | | 0.01–0.88 | 0.01–0.88 | |

Table 1 continued on next page

*Table 1 continued*

| Variables | First wave: March–April (MA) 2020 | | | Second wave: May–December (MD) 2020 | | |
|---|---|---|---|---|---|---|
| | Alive (N = 1683) | Dead (N = 423) | p-Value | Alive (N = 594) | Dead (N = 82) | p-Value |
| **PCR** | | | <0.001* | | | 0.004* |
| N-Miss | 47 | 12 | | 21 | 0 | |
| Mean (SD) | 77.25 (75.76) | 117.68 (95.97) | | 64.28 (73.38) | 98.59 (102.49) | |
| Median (Q1, Q3) | 55.65 (17.30, 111.60) | 99.20 (42.80, 170.45) | | 39.10 (12.30, 91.10) | 74.80 (20.12, 140.73) | |
| Range | 0.30–479.00 | 0.70–471.10 | | 0.30–483.20 | 0.30–593.80 | |
| **WBC** | | | <0.001* | | | 0.011* |
| N-Miss | 21 | 19 | | 4 | 1 | |
| Mean (SD) | 7.73 (7.13) | 9.13 (7.46) | | 7.65 (4.17) | 9.23 (6.25) | |
| Median (Q1, Q3) | 6.62 (4.87, 9.11) | 7.62 (5.60, 10.74) | | 6.67 (5.02, 8.90) | 8.34 (5.55, 12.04) | |
| Range | 0.72–191.02 | 0.32–92.23 | | 0.97–48.19 | 0.97–47.79 | |
| **Basophils** | | | 0.073* | | | 0.419* |
| N-Miss | 23 | 19 | | 4 | 1 | |
| Mean (SD) | 0.02 (0.02) | 0.02 (0.02) | | 0.02 (0.04) | 0.02 (0.02) | |
| Median (Q1, Q3) | 0.01 (0.01, 0.02) | 0.01 (0.01, 0.02) | | 0.02 (0.01, 0.03) | 0.01 (0.01, 0.03) | |
| Range | 0.00–0.31 | 0.00–0.15 | | 0.00–0.84 | 0.00–0.11 | |
| **Basophils %** | | | <0.001* | | | 0.024* |
| N-Miss | 22 | 19 | | 4 | 1 | |
| Mean (SD) | 0.00 (0.00) | 0.00 (0.00) | | 0.00 (0.00) | 0.00 (0.00) | |
| Median (Q1, Q3) | 0.00 (0.00, 0.00) | 0.00 (0.00, 0.00) | | 0.00 (0.00, 0.00) | 0.00 (0.00, 0.00) | |

*Table 1 continued on next page*

*Table 1 continued*

| Variables | First wave: March–April (MA) 2020 | | | Second wave: May–December (MD) 2020 | | |
|---|---|---|---|---|---|---|
| | Alive (N = 1683) | Dead (N = 423) | p-Value | Alive (N = 594) | Dead (N = 82) | p-Value |
| Range | 0.00–0.02 | 0.00–0.06 | | 0.00–0.05 | 0.00–0.01 | |
| **Eosinophils** | | | *<0.001** | | | *0.015** |
| N-Miss | 23 | 19 | | 4 | 1 | |
| Mean (SD) | 0.06 (0.12) | 0.04 (0.10) | | 0.06 (0.14) | 0.05 (0.13) | |
| Median (Q1, Q3) | 0.01 (0.00, 0.07) | 0.00 (0.00, 0.02) | | 0.01 (0.00, 0.06) | 0.00 (0.00, 0.03) | |
| Range | 0.00–2.19 | 0.00–0.79 | | 0.00–1.95 | 0.00–0.97 | |
| **Eosinophils %** | | | *<0.001** | | | *0.013** |
| N-Miss | 22 | 19 | | 4 | 1 | |
| Mean (SD) | 0.01 (0.02) | 0.00 (0.01) | | 0.01 (0.02) | 0.01 (0.01) | |
| Median (Q1, Q3) | 0.00 (0.00, 0.01) | 0.00 (0.00, 0.00) | | 0.00 (0.00, 0.01) | 0.00 (0.00, 0.00) | |
| Range | 0.00–0.27 | 0.00–0.12 | | 0.00–0.25 | 0.00–0.07 | |
| **Monocytes** | | | *<0.001** | | | 0.683* |
| N-Miss | 23 | 19 | | 4 | 1 | |
| Mean (SD) | 0.56 (0.68) | 0.69 (3.32) | | 0.55 (0.32) | 0.58 (0.41) | |
| Median (Q1, Q3) | 0.47 (0.32, 0.68) | 0.41 (0.25, 0.63) | | 0.49 (0.33, 0.68) | 0.48 (0.27, 0.77) | |
| Range | 0.01–23.31 | 0.02–66.34 | | 0.02–2.45 | 0.07–2.01 | |
| **Monocytes %** | | | *<0.001** | | | *0.034** |
| N-Miss | 22 | 19 | | 4 | 1 | |
| Mean (SD) | 0.08 (0.04) | 0.07 (0.05) | | 0.08 (0.04) | 0.07 (0.05) | |

*Table 1 continued on next page*

 Biochemistry and Chemical Biology | Medicine

Table 1 continued

| Variables | First wave: March–April (MA) 2020 | | | Second wave: May–December (MD) 2020 | | |
|---|---|---|---|---|---|---|
| | Alive (N = 1683) | Dead (N = 423) | p-Value | Alive (N = 594) | Dead (N = 82) | p-Value |
| Median (Q1, Q3) | 0.07 (0.05, 0.10) | 0.06 (0.04, 0.08) | | 0.07 (0.05, 0.10) | 0.06 (0.04, 0.09) | |
| Range | 0.00–0.70 | 0.01–0.72 | | 0.01–0.31 | 0.01–0.27 | |
| Ferritin F | 613 patients (82.39%) | 131 patients (17.61%) | <0.001* | 240 patients (90.23%) | 26 patients (9.77%) | 0.372* |
| N-Miss | 158 | 34 | | 43 | 5 | |
| Mean (SD) | 674.53 (817.61) | 1237.07 (2308.64) | | 564.63 (526.39) | 2006.00 (4680.23) | |
| Median (Q1, Q3) | 459.00 (212.00, 820.50) | 700.00 (353.00, 1347.00) | | 433.00 (216.00, 750.00) | 510.00 (269.00, 722.00) | |
| Range | 4.00–7687.00 | 19.00–20,572.00 | | 11.00–3397.00 | 81.00–20,941.00 | |
| Ferritin M | 1070 patients (78.56%) | 292 patients (21.44%) | <0.001* | 354 patients (90.23%) | 56 patients (9.77%) | 0.007* |
| N-Miss | 257 | 96 | | 50 | 5 | |
| Mean (SD) | 1353.00 (1359.86) | 1825.25 (1945.47) | | 1181.95 (3295.92) | 1372.04 (1258.14) | |
| Median (Q1, Q3) | 939.00 (461.00, 1705.00) | 1262.50 (572.25, 2323.25) | | 737.50 (405.25, 1283.00) | 1159.00 (598.00, 1500.00) | |
| Range | 23.00–11,513.00 | 55.00–13,289.00 | | 25.00–56,039.00 | 112.00–7058.00 | |

In bold and italics p-values < 0.05.
*Wilcoxon rank-sum test.
†Fisher's exact test.

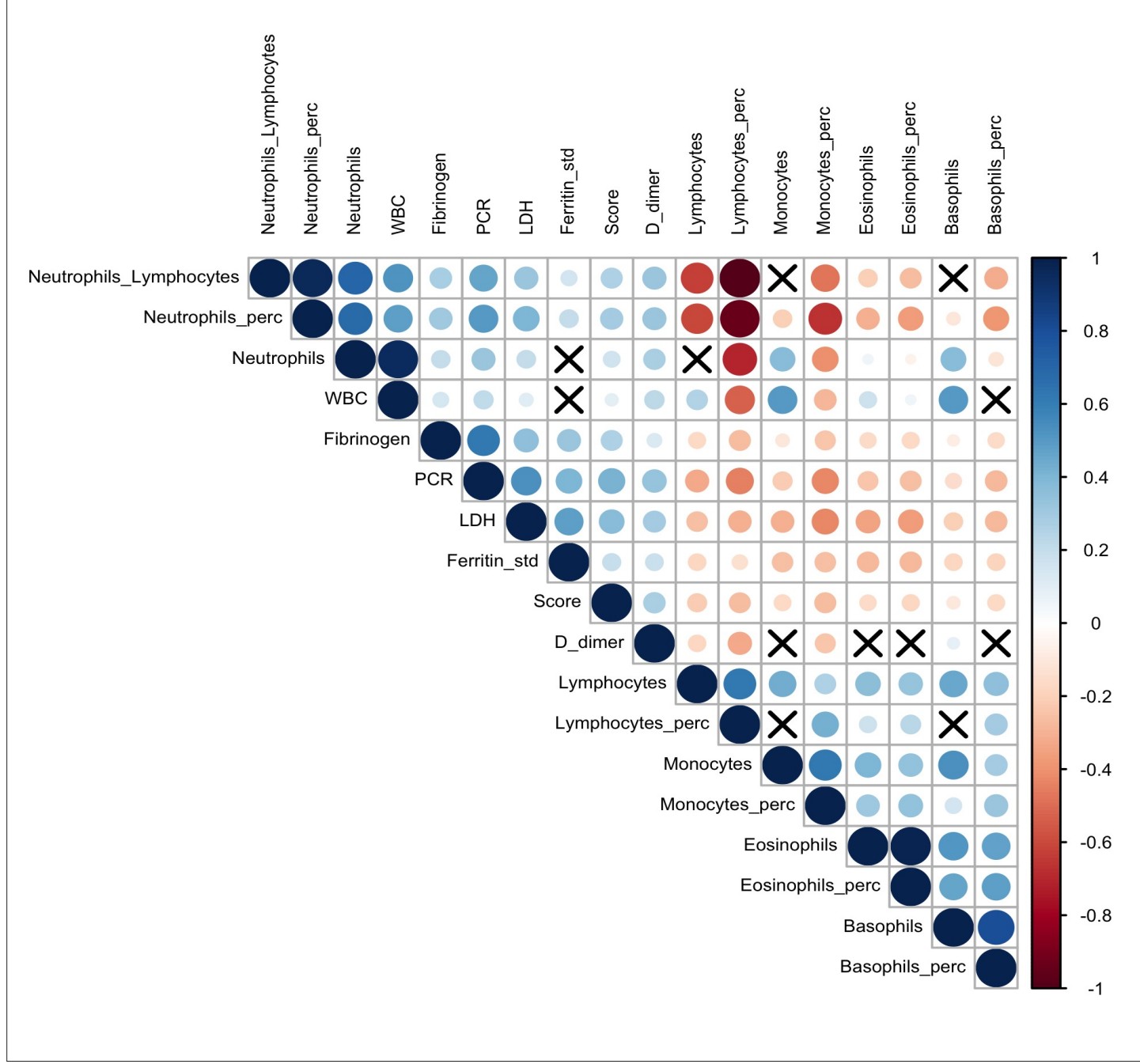

**Figure 2.** Correlation plot on biomarkers and Brescia chest X-ray score. The relationships between 17 analytes and Brescia chest X-ray score are inspected with the Spearman correlation coefficients, $\rho$ s, which are represented in this correlation plot by means of blue and red circles (positive and negative correlation, respectively). The diameter of the circle is proportional to the magnitude of $\rho$ s and black crosses on them identify correlation not significantly different from zero (p-values > 0.05). The correlation matrix is reordered according to the hierarchical cluster analysis on the quantitative variables.

## BS-EWM

A machine-learning model (BS-EWM) was developed by inputting a dataset of 2782 COVID-19 patients admitted to the ED and hospitalized at SCBH from March to December 2020. The majority of patients (2106/2782, 75.70%) belonged to the first wave (MA), the remaining fraction (676/2782, 24.30%) to the second wave (MD). As outcome, the machine-learning model had the condition dead/ alive, and, as covariates: age, Brescia X-ray score, and 17 blood sample analytes.

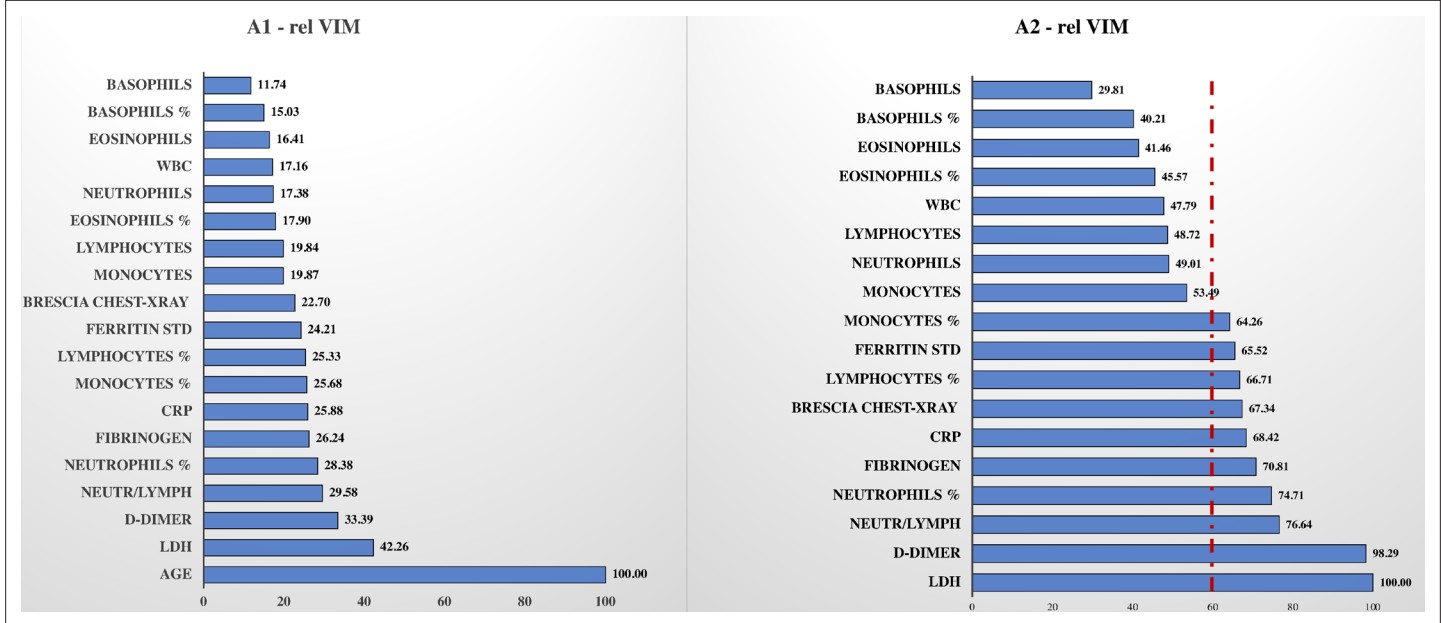

**Figure 3.** Relative variable importance measure (rel VIM). In , *Figure 3*, A1, there is the rel VIM based on Gini index. It was extracted from a random forest where the outcome is dead/alive and covariates are: the 17 biomarkers, Brescia X-ray score, and age. The algorithm grows 10,000 trees where the number of splitting variables at each tree node is √(# covariates in the model). Missing values are imputed with the 'on-the-fly-imputation' algorithm. A model with the same features was run (*Figure 3*, A2) excluding the covariate 'age' since it was strongly associated with the risk of death, masking the role of remaining covariates.

*Figure 1* reports the flowchart that describes how data were divided for training, validation, and testing the BS-EWM.

The synthetic minority oversampling technique (SMOTE) procedure, rebalancing the dead/alive ratio (50% vs. 50%) from the original 20.09%, improved accuracy, specificity, and sensitivity of the random forest applied on it (see *Supplementary file 1c* which compares performance metrics with/ without the SMOTE method).

The relative variable importance measure (rel VIM) and partial dependence plot (PDP) were extracted from the random forests (*Figures 3 and 4*, respectively). In *Figure 3*, the rel VIM of BS-EWM based on age, Brescia X-ray score, and 17 blood analytes are reported on a bar plot. Since age was strongly associated with the risk of death, it masked the role of the other covariates. For completeness, the relevance of the 17 analytes and Brescia X-ray score was estimated in an additional EWM, in which the covariate 'age' was excluded. In *Figure 4*, 9/17 analytes and the Brescia X-ray score were noted as being important in predicting the risk of death (rel VIM >60). The effects of changes in covariate values on the risk of death threshold of the EWM were reported by means of a PDP (a 2D plot in the x–y plane) (*Figure 4*). Only fibrinogen was excluded from this graphical representation since in *Table 1*, it was not significantly different in the two subpopulations deceased/alive. Most PDPs showed nonmonotonic increasing relationships between the x-variable and the EWM, resulting in a plateau corresponding to high values of x.

When compared to other models such as gradient boosting machine (GBM) and logistic regression, the random forest showed better performance in terms of area under the curve (AUC), sensitivity, and specificity. The in-sample sensitivity (0.93) yielded by the model was the highest, and it maintained an important 0.82 in validating the out-of-sample sensitivity, and this decreased to 0.73 when testing the MD subgroup (see *Table 2* which contains details on all the metrics extracted from the ROC analysis). ROC curves are visualized in *Figure 5* where, for each model (random forest, GBM, and logistic regression), the performances in training, validating, and testing are compared in a unique graph.

In order to compare the BS-EWM score with univariate models based on single biomarkers, three random forest (on training, validation, and testing) are estimated on the most important biomarkers (LDH, D-dimer, neutr/lymph, neutrophils %, fibrinogen, CRP, Brescia chest X-ray, lymphocytes %, ferritin std, monocytes %). Results of these 30 models are reported in *Supplementary file 1d*. It is

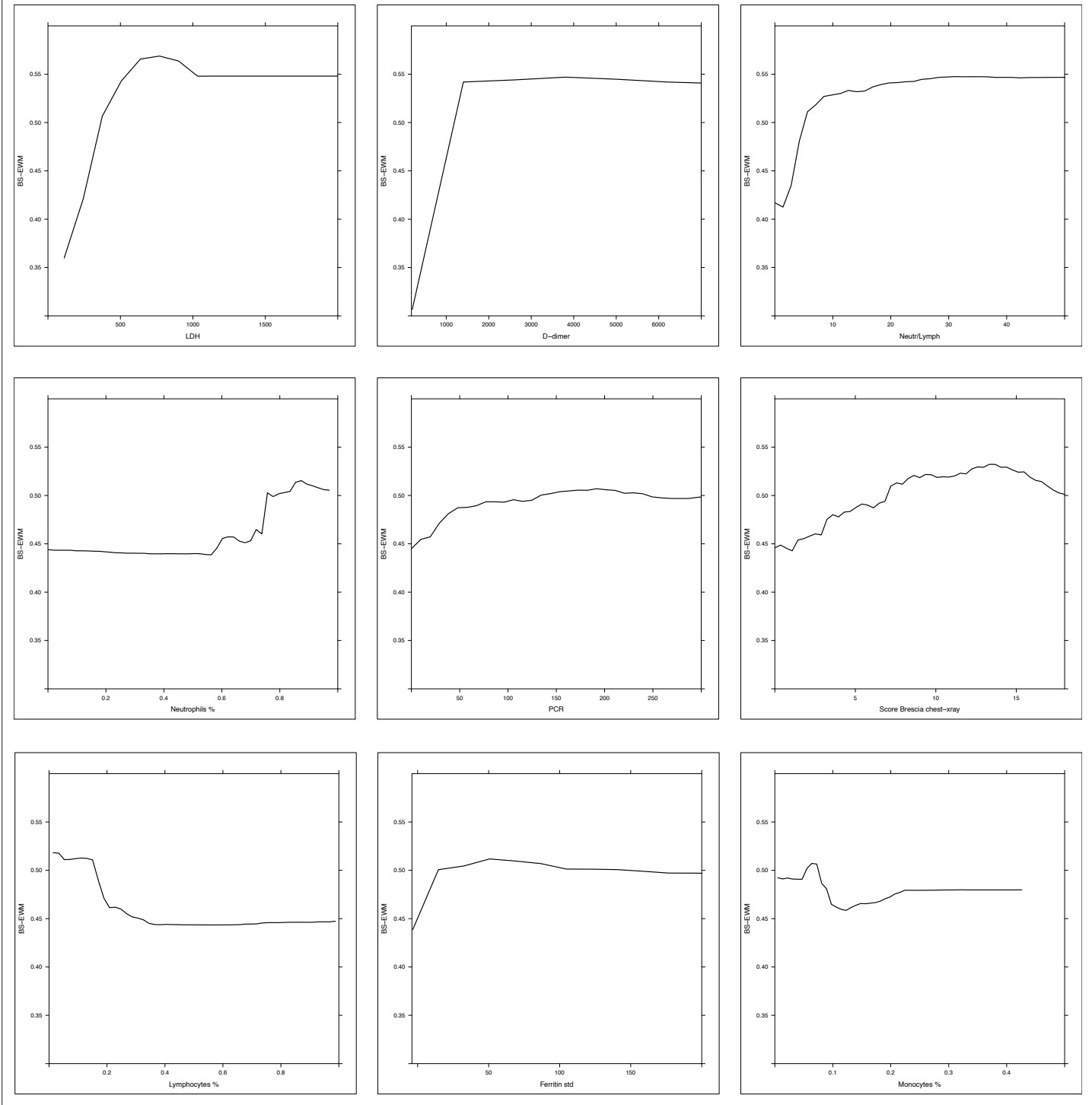

**Figure 4.** Partial dependence plot (PDP) of random forest grown on the 17 biomarkers and Brescia X-ray score. Considering the random forest that excludes the 'age' variable, the PDPs were computed in correspondence of covariates with relative variable importance measure (rel VIM) of Appendix 1—figure 2 > 60 (cut-off identified by the red dashed line) and p-value in *Table 1* < 0.05. Of 10 most important variables in Appendix 1—figure 2, nine satisfy these two conditions (only fibrinogen was excluded since it was not significantly different in the two subpopulations deceased/alive). PDPs measure the effects of changes in covariate values taken one per time, on the risk of death. They are displayed from the most to the less important variable.

**Table 2.** Performance metrics of methods: random forest, gradient boosting machine (GBM), and logistic regression.

| Metrics | Random forest | | | GBM | | | Logistic regression | | |
|---|---|---|---|---|---|---|---|---|---|
| | Training March–April (MA) | Validating March–April (MA) | Testing May–Dec (MD) | Training March–April (MA) | Validating March–April (MA) | Testing May–Dec (MD) | Training March–April (MA) | Validating March–April (MA) | Testing May–Dec (MD) |
| AUC (DeLong) (95% CI) | 0.97 (0.97–0.98) | 0.83 (0.80–0.87) | 0.78 (0.73–0.84) | 0.88 (0.86–0.89) | 0.84 (0.80–0.88) | 0.78 (0.73–0.83) | 0.84 (0.82–0.86) | 0.83 (0.79–0.87) | 0.52 (0.44–0.60) |
| Sensitivity (95% CI) | 0.93 (0.91–0.97) | 0.82 (0.72–0.92) | 0.73 (0.54–1.00) | 0.85 (0.80–0.88) | 0.80 (0.66–0.90) | 0.77 (0.65–0.94) | 0.80 (0.77–0.84) | 0.84 (0.76–0.91) | 0.87 (0.18–1.00) |
| Specificity (95% CI) | 0.92 (0.88–0.94) | 0.75 (0.63–0.83) | 0.73 (0.41–0.89) | 0.77 (0.73–0.81) | 0.75 (0.65–0.87) | 0.71 (0.50–0.79) | 0.74 (0.70–0.77) | 0.73 (0.65–0.79) | 0.26 (0.11–0.94) |

Comparison between the performances of three methods: random forest, GBM, and logistic regression model applied on the rebalanced dataset obtained with SMOTE methodology. Logistic regression predictions are computed using the 10-fold cross-validation in order to be comparable with random forest and GBM predictions (which use out-of-bag and 10-fold cross-validation, respectively).

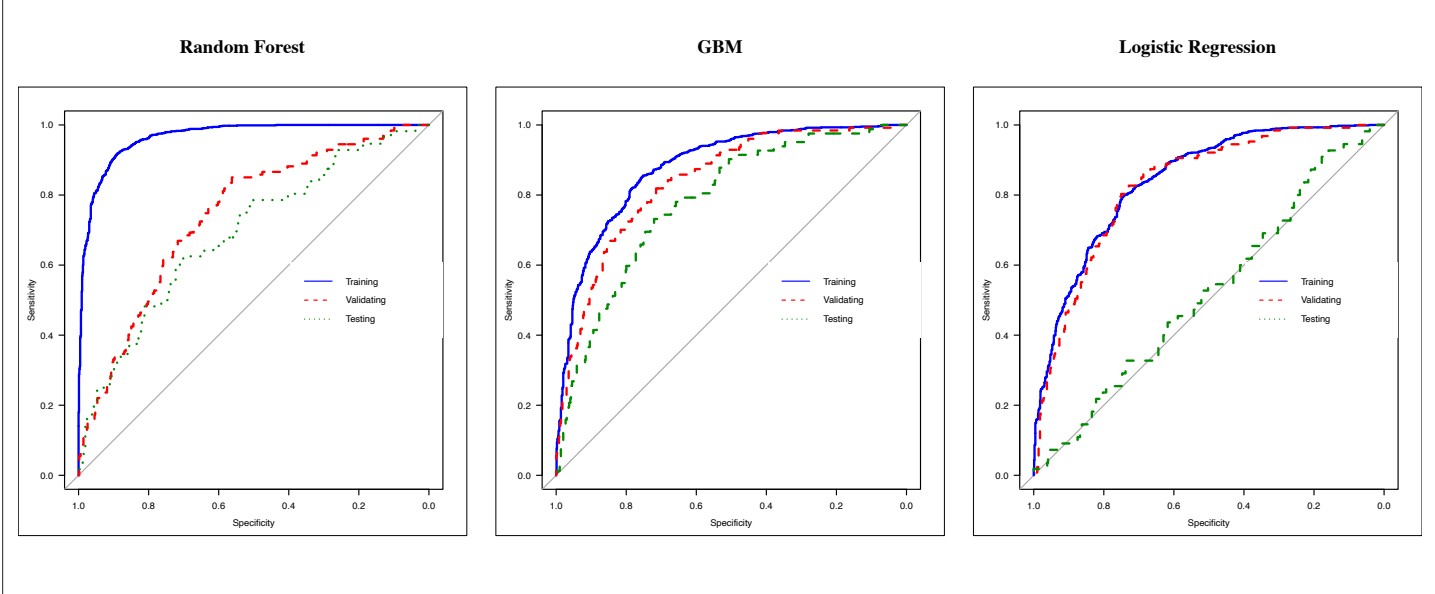

**Figure 5.** Receiver operating characteristic (ROC) curves of random forest, gradient boosting machine (GBM), and logistic regression. ROC curves of three methods: (i) random forest, (ii) GBM, and (iii) logistic regression. Each graph reports the ROC curve computed in training (blue line, 70 % of March–April's patients), validating (dashed red line, 30 % of March–April's patients), and testing (dashed green lined, May–December's patients).

evident that considering one biomarker per time, the model provides good predictions in training, but bad performances out of sample (contrary to the BS-EWM score it loses its predictive power on fresh data). Hence, it is evident that a score based on a multivariate model provide better results since it also considers interactions among variables.

## Discussion

The dataset for the development, validation, and testing of the BS-EWM originated entirely from an Italian region, potentially limiting the generalizability of the risk score in other areas of the world. Additional validation studies from different geographic areas are welcomed. Furthermore, though the BS-EWM has been validated using blood sample values obtained by instruments that satisfy internal and external quality control, different equipment could lead to divergent results (*Martens et al., 2021*; *Lippi et al., 2020*). Therefore, it would be appropriate to harmonize the results. Another limit could have been the presence of missing values, though the BS-EWM has also performed adequately in this condition since it used a multiple imputation technique to overcome the problem. Finally, it is important to point out that the BS-EWM risk score should not be used for asymptomatic COVID-19 patients or for the pediatric population. It will be interesting, in the future, to verify if the BS-EWM could be applied by general practitioners to the unhospitalized population. This could allow the generalizability of the model outside the hospital context to be tested.

Though the BS-EWM has been developed on a cohort of 2106 patients belonging to the COVID-19 first wave, the model also demonstrated a sensitivity greater than 70 % in the early prediction of high risk in patients in the second wave, when in-hospital mortality was 40 % lower.

Several predictive models have recently been applied to COVID-19 cohorts with variable results, some of them previously developed to predict mortality for community-acquired pneumonia, such as the Pneumonia Severity Index, CURB-65, qSOFA, and MuLBSTA (*Genc Yavuz et al., 2021*; *Lazar Neto et al., 2021*; *Artero et al., 2021*), NEWS2 criteria (*Myrstad et al., 2020*; *Gidari et al., 2020*), and SCAP score (Anurag and Preetam, n.d.). Novel early-warning scores have been specifically built on COVID-19 patient series using different techniques such as parametric and non-parametric tests (*Linssen et al., 2020*) or artificial intelligence techniques such as the COVID-GRAM score (*Liang et al., 2020*).

While these models are mostly based on age and a set of vital (clinical) parameters, in addition to age, the BS-EWM depends on blood parameters. It is conceivable that blood analytes capture a

snapshot at hospital admission signaling a specific bodily reaction to viral infection in terms of hyper-inflammation, immune response, and thrombophilia. On the other hand, the other models are more influenced by the general status of the patient, which may be determined by concomitant and pre-existing diseases.

According to the International Federation of Clinical Chemistry (*Bohn et al., 2020*), no single biochemical or hematological marker is sufficiently sensitive or specific to predict the outcome of SARS-CoV-2 infection. Notably, the IFCC recommends that the interpretation of laboratory abnormalities should be based on groups of analytes (*Bohn et al., 2020*). In the BS-EWM, three analytes reached a significant value in predicting death: LDH, D-dimer, and NLR. LDH is a non-specific indicator of tissue damage (*Bohn et al., 2020*; *Liang et al., 2020*) that emerges as one of the most consistently elevated markers in patients at higher risk of developing adverse outcomes, probably because COVID-19 infection is characterized by systemic tissue damage. Another key feature of SARS-CoV-2 is the coagulopathy: high levels of D-dimers have been reported to correlate with unfavorable disease progression in several cohorts of patients. The coagulopathy linked to COVID-19 infection is likely to involve a complex interplay between pro-thrombotic and inflammatory factors, thus the combined analysis of both inflammatory and thrombophilic markers could play an important role in the early identification of patients at higher risk of unfavorable progression (*Bohn et al., 2020*; *Lazzaroni et al., 2021*). Finally, lymphopenia has become a hallmark of SARS-CoV-2. It has been demonstrated in almost all symptomatic patients, though in varying degrees. Disease severity has been correlated with the level of lymphocyte count reduction. A direct infection of lymphocytes, which express the coronavirus receptor ACE-2, is among the mechanisms proposed. A poor prognosis is also associated with an elevated neutrophil count combined with lymphopenia, resulting in a high NLR. The increase in granulocytes is the result of the cytokine storm induced by the virus and is responsible for tissue damage (*Bonetti et al., 2020*; *Bohn et al., 2020*).

A further remark concerning the blood analytes is that, in the BS-EWM, the thresholds of the single analytes (namely the point where the functions in *Figure 4* become constant and the probability of death no longer increases/decreases) closely overlap with the values recently proposed by other authors (*Webb et al., 2020*; *Caricchio et al., 2021*). For completeness, the optimal threshold (computed through the Youden index) for each biomarker to predict the outcome (dead or alive) are reported in *Supplementary file 1e*.

The present study is not unique in encompassing radiological findings combined with blood analysis. The study by *Schalekamp et al., 2021*, integrated blood analysis parameters and radiological information derived by grading chest X-rays (0–8 scale points). Unlike the cited study, with the BS-EWM in this study, the radiological score did not reach a high relevance (rel VIM) in predicting high risk. This difference can be explained by the different approaches used to build the model (logistic regression vs. random forests) and by the high degree of correlation of the X-ray score with multiple blood analytes: 'collinearity' thus could have 'stolen' importance from the information provided by imaging. Nevertheless, at admission, the chest X-ray score of patients who subsequently died was significantly higher than for patients who survived. Furthermore, the chest X-ray score may provide additional stability to the model, playing an important role in the case of missing data in the blood sample counterpart.

Further, the BS-EWM delivers high prediction performance and only requires a limited number of readily available variables with easy operability, no time consuming, no extra money since these analytes are required for COVID-19 diagnosis and monitoring. An important and pragmatic aspect offered by the BS-EWM is that the biomarkers employed may be obtained by the emergency laboratory in less than an hour (*Garrafa et al., 2020a*) and, differently from other biomarkers (*Kyriazopoulou et al., 2021*), they are non-expensive and frequently used also in developing countries. It is important to note that the same methodology could be applied to other infections and be practical to triage people.

Most laboratories, including the small or peripheral ones, may provide results in a short time. At the Spedali Civili of Brescia, the BS-EWM is integrated within the Laboratory Information System (LIS). It works as a web-based calculator and is easy to interpret. The online calculator allows an easy assessment of the EWM, requiring only the entering of analyte values and X-ray score. The score is calculated even if some of the values are missing. Furthermore, in our center the system may be integrated with the electronic health record and the radiology information system, allowing a completely

automatic data retrieval and entering, without any operator interaction. It provides a risk threshold of 0.5, above which patients are graded as having a potentially high death-risk, thus supporting closer clinical observation or admission to a high-intensive care ward. In patients yielding a low risk (score 0–0.49), the decision by clinicians to allocate them to a low-intensive care ward or to monitoring is further sustained. The online calculator allows an easy assessment of the EWM, requiring only the entering of analyte values and X-ray score. The score is calculated even if some of the values are missing. Furthermore, in our center the system may be integrated with the electronic health record and the radiology information system, allowing a completely automatic data retrieval and entering, without any operator interaction.

Finally, the need to regularly update models and closely monitor their performances over time and geographically should be underlined, given the rapidly changing nature of the disease and its management.

## Materials and methods

The dataset contained 2782 COVID-19 symptomatic patients, hospitalized between March and December 2020 at SCBH after referring to the ED. In all patients, the following variables were retrieved from the SCBH database: age, sex, length of hospitalization, Brescia X-ray score (*Borghesi and Maroldi, 2020*), alive/dead, and 17 blood analytes acquired at admission (D-dimer, fibrinogen, LDH, neutrophils, lymphocytes, NLR, lymphocytes %, neutrophils %, CRP, white blood cell (WBC) count, basophils, basophils %, eosinophils, eosinophils %, monocytes, monocytes %, standardized ferritin). Blood tests were acquired within 24 hr after admission to the hospital.

According to the two temporal peaks of incidence of the COVID-19 outbreak in Lombardy, the 2782 patients were divided into two groups: (i) MA including 2106 patients admitted during the first wave; (ii) MD including 676 patients in the second wave. Quantitative variables were described using mean (SD), median (IQR), and range (min–max), while categorical variables were reported as counts and percentages. The comparisons between groups were performed using the Wilcoxon rank-sum test for quantitative variables and Fisher's exact test for qualitative variables.

The relationships between the 17 analytes and the Brescia X-ray score were inspected using the Spearman correlation coefficient, $\rho$ s, and visualizing results using a correlation plot (*Dancelli et al., 2013*; *Marziano et al., 2019*; *Figure 2*).

To estimate the BS-EWM, the outcome (alive/dead) was modeled using as covariates: (i) Brescia X-ray score, (ii) 17 analytes, (iii) age. Since most of the covariates analyzed were strongly correlated (multi-collinearity) (*Figure 2*) and their relationships with the outcome were non-linear, the BS-EWM was estimated using random forests (*Breiman, 2001*; *Carpita and Vezzoli, 2012*), a non-parametric machine-learning method (*Vezzoli, 2011*; *Vezzoli et al., 2017*). Moreover, the algorithm is able to manage missing values which are common in clinical studies. The 'on-the-fly-imputation' algorithm (*Hong and Lynn, 2020*) imputes data when it grows the forest handling interactions and non-linearity in the dataset.

Since the prevalence rate of death in the two waves was different (20 % in MA vs. 12 % in MD), a strategy to generalize results in unbalanced datasets was applied, adopting a rebalancing method able to improve the detection of patients with a high death-risk.

The EWM was developed using the 2106 patients in the first COVID-19 wave (MA 2020) when in-hospital death prevalence was 20 %. Seventy percent of them (1474 patients) were used for training the model and the remainder (632 patients) for testing it. Patients were randomly assigned to the two subgroups, and further stratified according to the outcome (alive/dead). Consequently, both the training and testing subgroups included the same rate of deaths (20.09%) as the full sample (2106 patients). With such a 'moderate' incidence of death, the dataset was statistically unbalanced. This limitation could have implied the development of a model yielding unsatisfactory results in predicting new observations for the minority class, that is, patients with death as outcome. An approach to address this limitation is to oversample the minority class (deceased patients) and, subsequently, create the predictive model (BS-EWM). The SMOTE (*Chawla et al., 2002*) was chosen. The SMOTE function oversamples the minority class by using bootstrapping and k-nearest neighbor to synthetically create additional observations belonging to that class (dead). The procedure is combined with under-sampling of the majority class (alive). To determine the optimum number of k-groups into which to assign the dataset, a matrix containing the 17 analytes and the Brescia X-ray score was used

to compute the hierarchical cluster analysis (*Salvi et al., 2019*; *Codenotti et al., 2016*). By means of silhouette analysis, k = 2 was determined as the optimal number of clusters into which to assign the dataset. Hence, a synthetic rebalanced dataset was obtained with an equal number of living and deceased patients (888 + 888). The rebalancing procedure enabled a risk score to be devised ranging from 0 to 1 with a threshold of 0.5 to separate non-severely affected from severely affected patients. Subsequently, the model was tested on the subgroup of 632 patients in the first wave excluding the training set. A further validation of the EWM was conducted on the 676 COVID-19 patients in the second wave (*Wynants et al., 2020*).

The rel VIM (*Carpita and Vezzoli, 2012*, *Doglietto et al., 2020b*) and the PDP (*Friedman, 2001*; *Doglietto et al., 2020a*) were extracted from the model for a better understanding of the relationship between outcome and covariates.

The predictions extracted from the random forests classification were interpreted as in-hospital death probability conditional on the combination of the values of analytes, Brescia X-ray score, and age in COVID-19 patients at admission to the ED.

The BS-EWM performance was evaluated by AUC of an ROC curve. The robustness of the model was compared to other models by running GBM (a machine-learning approach and competitor to random forests), and logistic regression, and computing the same metrics.

The BS-EWM score is available for use online (https://bdbiomed.shinyapps.io/covid19score). In the SCBH it is integrated within LIS returning the death-risk score directly from the medical report.

All the analyses were performed by R, version 4.0.0 (*R Development Core Team, 2020*). The code is available at 288 https://github.com/biostatUniBS/BS_EWS copy archived at swh:1:rev:7416ba71075402e6a0ed997e7aa6a527e93247b2 (*Garrafa et al., 2021*).

## Additional information

### Funding

| Funder | Grant reference number | Author |
|---|---|---|
| Italian Ministry of University and Research | PRIN 20178S4EK9 | Stefano Calza |

The funders had no role in study design, data collection and interpretation, or the decision to submit the work for publication.

### Author contributions

Emirena Garrafa, Conceptualization, Data curation, Investigation, Methodology, Resources, Supervision, Validation, Writing – original draft, Writing – review and editing; Marika Vezzoli, Conceptualization, Data curation, Formal analysis, Investigation, Methodology, Software, Supervision, Writing – original draft, Writing – review and editing; Marco Ravanelli, Conceptualization, Data curation, Supervision, Writing – review and editing; Davide Farina, Supervision, Writing – review and editing; Andrea Borghesi, Validation, Writing – review and editing; Stefano Calza, Conceptualization, Data curation, Funding acquisition, Software, Supervision, Writing – review and editing; Roberto Maroldi, Conceptualization, Data curation, Supervision, Writing – original draft, Writing – review and editing

### Author ORCIDs

Emirena Garrafa http://orcid.org/0000-0003-4761-6892
Marco Ravanelli http://orcid.org/0000-0002-4841-2771

### Ethics

Human subjects: The Institutional review board approved the study with the entry code NP4000.

### Decision letter and Author response

Decision letter https://doi.org/10.7554/eLife.70640.sa1
Author response https://doi.org/10.7554/eLife.70640.sa2

## Additional files

### Supplementary files

• Supplementary file 1. Descriptive statistics on all variables. (a) Descriptive statistics on all variables of the entire sample. (b) Descriptive statistics on all variables in the dataset stratified respect first (March–April 2020) and second (May–December 2020) wave. Comparison between alive and dead. (c): Performance metrics of the random forest (RF) using or not a rebalanced dataset with the synthetic minority oversampling technique (SMOTE) methodology. In this table we compare the performance of two RFs applied on (i) a dataset rebalanced with the SMOTE methodology and (ii) the original dataset. This analysis suggests the use of SMOTE methodology before applying RF since the performance in training and validating groups (especially in terms of sensitivity) are better respect those obtained from the RF grown on the original dataset. (d): Performance metrics of the random forests (RFs) estimated on single biomarkers. (e): Optimal threshold for each biomarker to predict the outcome.

### Data availability

We are unable to share the dataset as it contains sensitive personal data collected during the pandemic at Spedali Civili di Brescia. We cannot share the full data since are data from patients. Interested researchers should contact the authors for any query related to data sharing and submit a project proposal Once defined the goal of the study, and the data needed authors will submit the potential project of collaboration to the IRB of Spedali Civili di Brescia to receive approval to access a deidentified dataset. Please note that other informations related to patients can be acquired, always following approval of IRB of Spedali Civili di Brescia, not only the ones studied in the paper. Anyway, following request to the authors, it will be possible to share processed version of the dataset ( e.g. an Excel sheet with numbers used to plot the graphs and charts of the manuscript). All code used to analyse the data can be found on GitHub at https://github.com/biostatUniBS/BS_EWS, (copy archived at swh:1:rev:7416ba71075402e6a0ed997e7aa6a527e93247b2).

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
