## [Decision Letter]

**Acceptance summary:**

Garrafa et al., developed and validated an early-warning model in order to predict in-hospital mortality in patients affected by COVID-19 during the first and second wave of the pandemic. The model includes age, blood analytes and a chest X-ray score, all parameters that are easy to get at the emergency room admission. The authors tested the model in a training, validating and testing sample, showing an AUC of 0.98, 0.83 and 0.78, respectively. One of the major strengths of the study is the large sample collected in one of the main hospital of Northern Italy as well as the amount of data collected regarding analytes and radiological findings. The statistical analysis is appropriate and detailed.

**Decision letter after peer review:**

Thank you for submitting your article "Early Prediction of In-Hospital Death of COVID-19 Patients: A Machine-Learning Model Based on Age, Blood Analyses, and Chest X-Ray Score" for consideration by *eLife*. Your article has been reviewed by 3 peer reviewers, and the evaluation has been overseen by a Reviewing Editor and Jos Van der Meer as the Senior Editor. The following individuals involved in review of your submission have agreed to reveal their identity: Alessandra Marengoni (Reviewer #1); Dimitrios Velissaris (Reviewer #2).

Essential revisions:

– The authors may want to add the following reference: doi: 10.1136/bmj.m3339

– The predictive value of the BS-EWM could be evaluated with other scores such as the COVID-19 Mortality Score, COVID-19 Severity Index, 4C Mortality Score and COVID-IRS NLR. Do the authors have the data to provide comparisons with these scores?

– The authors describe that they used parameters of patients when admitted in the ED but intended to report in-hospital mortality. I suppose a large number of patients weren't admitted in hospital and those data would be of interest for validity of the score.

– The authors do not describe the time of symptoms of patients. This would be also interesting to consider alongside with the score as X-ray findings change over time.

– It would be interesting to see data of comparison of the predictive value of well-established single severity biomarkers (LDH, lymphocytes, d-dimer etc) with the score.

– A comment about the correlation between the X.ray score and the analytes should be added in the discussion.

– Line 231-233; the authors may want to expand the paragraph comparing thresholds of analytes in other studies.

– Also, the authors should expand in the discussion the easy-to-use online score.

---

## [Author Response]

Essential revisions:– The authors may want to add the following reference: doi: 10.1136/bmj.m3339

Many thanks for your suggestion. We added the citation in the bibliography and in the article (line 55).

– The predictive value of the BS-EWM could be evaluated with other scores such as the COVID-19 Mortality Score, COVID-19 Severity Index, 4C Mortality Score and COVID-IRS NLR. Do the authors have the data to provide comparisons with these scores?

Thanks for your comment. We do not have available all the data needed to calculate those other scores. In particular we lack clinical data about symptoms and comorbidity which are collected by physicians during the medical examination. This is a limitation of our work given the unquestionable impact of this data on patient prognosis. Otherwise, our score, relying only on laboratory and instrumental numeric data, has the advantage to be automatically assessable without operator interaction. This makes it appealing in a context like ours where the initial triage of patients is not performed by expert physicians but for the most part by other sanitary personnel. In this way, the score may become part of a workflow in which the patient has already a risk profile when arriving to the doctor’s visit. This represents a ready-to-use tool for the examining physician who can integrate it with clinical data and make the most proper decision for patient managing.

– The authors describe that they used parameters of patients when admitted in the ED but intended to report in-hospital mortality. I suppose a large number of patients weren't admitted in hospital and those data would be of interest for validity of the score.

Thanks for highlighting this aspect. Actually, our model has not been validated for those patients who were not admitted in hospital. This is due to the lack of information about the outcome of this considerable part of patients. We do not have the possibility to retrieve these data and for this we will enhance this aspect in the limitation section of discussion. If our score will become public, it could be applied by general practitioners to the unhospitalized population. This could allow the generalizability of the model outside the hospital context to be tested (line 140-142)*.*

– The authors do not describe the time of symptoms of patients. This would be also interesting to consider alongside with the score as X-ray findings change over time.

We don’t have available data about the onset and the variation of symptoms over time. Regarding the change of X-ray during the hospitalization, data are available and already published for a significant subsample of the first wave patients (doi: 10.1007/s00330-020-07504-2). However, it is important to underline that the analysis of evolutive changes in X-ray score and analytes is not the aim of our study, given that we intended to create an early predictive model based only on parameters at admission.

– It would be interesting to see data of comparison of the predictive value of well-established single severity biomarkers (LDH, lymphocytes, d-dimer etc) with the score.

In order to replay to this point, we have selected the ten most important biomarkers (LDH, D-dimer, Neutr/Lymph, Neutrophils %, Fibrinogen, CRP, Brescia chest xray, Lymphocites %, Ferritin std, Monocytes %) and, for each of them, we estimated three RF: on training set (March-April, MA), on validating (March-April, MA) and on testing (May-December, MD). Results of these 30 models are reported in the Table Supplementary file 1d. It is evident that considering one biomarker per time, it provides good predictions in training, but bad performances out of sample (it loses its predictive power on fresh data). On the contrary, our score based on 19 biomarkers, provides good results in sample and out of sample since it considers the jointly role and the interactions of the variables involved.

Since these results are very interesting and reinforce the predictive ability of our score, we add the table in Supplementary Materials (Table Supplementary file 1d). The corresponding comment is reported in lines 122-129 of the revised paper.

– A comment about the correlation between the X.ray score and the analytes should be added in the discussion.

Thanks for your suggestion. The correlation between x-ray score and analyte values is represented in the figure 2. We observed a strong positive correlation between the x-ray score and some analytes, in particular LDH, CRP and neutrophil percentage, which all are markers of an inflammatory status. Chest x-ray is performed in order to stage the pulmonary involvement in patients with a biohumoral status of inflammation.

– Line 231-233; the authors may want to expand the paragraph comparing thresholds of analytes in other studies.

Since this concept was not well explained, we modify sentence in line 181-186. For completeness, we computed also the optimal threshold (by means of Youden index) for each biomarker to predict the outcome (dead or alive). We finally add the table in Supplementary Materials (Table S.5).

– Also, the authors should expand in the discussion the easy-to-use online score.

Thanks to your suggestion. We expand discussion starting (line 198-205). The online calculator allows an easy assessment of the EWM, requiring only the entering of analyte values and x-ray score. The score is calculated even if some of the values are missing. Furthermore, in our center the system may be integrated with the electronic health record and the radiology information system, allowing a completely automatic data retrieval and entering, without any operator interaction. In addition, pushing forward with the system automation, we described in a recent publication an end-to-end artificial-intelligence system able to calculate the x-ray score from radiological images with single click.